# Carboxymethyl Cellulose-Based Polymers as Promising Matrices for Ficin Immobilization

**DOI:** 10.3390/polym15030649

**Published:** 2023-01-27

**Authors:** Andrey V. Sorokin, Svetlana S. Goncharova, Maria S. Lavlinskaya, Marina G. Holyavka, Dzhigangir A. Faizullin, Maxim S. Kondratyev, Sergey V. Kannykin, Yuriy F. Zuev, Valeriy G. Artyukhov

**Affiliations:** 1Biophysics and Biotechnology Department, Voronezh State University, 1 Universitetskaya Square, 394018 Voronezh, Russia; 2Laboratory of Bioresource Potential of Coastal Area, Institute for Advanced Studies, Sevastopol State University, 33 Studencheskaya Street, 299053 Sevastopol, Russia; 3Laboratory of Metagenomics and Food Biotechnologies, Voronezh State University of Engineering Technologies, 19 Revolutsii Avenue, 394036 Voronezh, Russia; 4Kazan Institute of Biochemistry and Biophysics, FRC Kazan Scientific Center of the RAS, 2/31 Lobachevsky Street, 420111 Kazan, Russia; 5Laboratory of Structure and Dynamics of Biomolecular Systems, Institute of Cell Biophysics of the RAS, 3 Institutskaya Street, 142290 Pushchino, Russia; 6Material Science and Nanosystem Industry Department, Voronezh State University, 1 Universitetskaya Square, 394018 Voronezh, Russia

**Keywords:** ficin, immobilization, carboxymethyl cellulose, graft copolymer, proteolytic activity

## Abstract

The present work is devoted to research on the interaction between carboxymethyl cellulose sodium salt and its derivatives (graft copolymer of carboxymethyl cellulose sodium salt and *N*,*N*-dimethyl aminoethyl methacrylate) with cysteine protease (ficin). The interaction was studied by FTIR and by flexible molecular docking, which have shown the conjugates’ formation with both matrices. The proteolytic activity assay performed with azocasein demonstrated that the specific activities of all immobilized ficin samples are higher in comparison with those of the native enzyme. This is due to the modulation of the conformation of ficin globule and of the enzyme active site by weak physical interactions involving catalytically valuable amino acids. The results obtained can extend the practical use of ficin in biomedicine and biotechnology.

## 1. Introduction

The cellulosic-derived materials and substances are abundant and frequently applied in human activity due to cellulose, the most common biopolymer in the world presented in plants, microorganisms, and algae [1,2,3]. Natural cellulose is a fibrous linear non-ionogenic, and non-water-soluble polymer. However, its physicochemical properties, including solubility and reactivity, can be tuned by chemical modification such as carboxymethylation, turning it to carboxymethyl cellulose sodium salt (Na-CMC), which is a swellable and water-solvable polyanion [4,5]. This modified polymer is manufactured industrially, being rather cheap and readily available. Moreover, the carboxymethyl cellulose sodium salt forms high viscosity solutions and elastic porous films, which are non-immunogenic and biocompatible [6,7]. These make it promising for the production of drugs and bioactive substance carriers, food thickeners, superabsorbing or wound-healing materials, etc. 

The repeating unit of carboxymethyl cellulose sodium salt contains up to three carboxylic or hydroxy groups in *D*-pyranose cycles which are linked to each other by 1,4-β-glycosidic bonds. The rigid fibrous polyanion structure of the Na-CMC macromolecules can negatively affect the application of the polymer [8]. To enhance it, further chemical modification is necessary. Graft polymerization, which is the production of side macromolecular substitutes with desirable properties, is a simple and effective method of property-improving polysaccharide modification [9,10,11]. New functional group incorporation into Na-CMC may raise the complexing ability of the resulting polymer. Moreover, it was shown that the grafting of poly(*N*-vinylimidazole) or poly(*N*,*N*-dimethyl amino ethyl methacrylate) (PDMAEMA) on Na-CMC macromolecules increases the loading and encapsulation efficiency of paclitaxel as compared to non-modified Na-CMC [12,13]. Paclitaxel is a plant-origin drug with a complex structure [14], so it can be suggested that incorporation of the side nitrogen-containing macromolecular substitutes may increase the interactions with other complex biostructures, e.g., enzymes [15].

Cysteine proteases have gained much attention in the research and industrial field. Some of them, for example, ficin, are characterized by their broad substrate specificity, pH (4.46–9), and temperature optimum (30–60 °C) and are used as antibiofilm agents and biocatalyzers in meat tenderization [16,17]. Moreover, the plant-origin ficin (EC 3.4.22.3) is isolated from the latex of genus *Ficus* [18], so it is renewable and non-expensive as compared to animal-origin enzymes. The ficin active site includes cysteine residue whose thiol group acts as a nucleophile in substrate hydrolysis. However, the SH-group is very sensitive toward external factors, so ficin is low-stable in solutions [19]. The most common way to protect the enzyme’s active site and enhance its stability is ficin immobilization on reactive functional polymers [20]. Since ficin is widely used in biomedicine and biotechnology, the immobilization supports must be non-toxic, non-immunogenic, biocompatible, and cheap [21]. So, carboxymethyl cellulose sodium salt and its derivatives are excellently fitted for this purpose. Additionally, there are numerous examples in which cellulose-based materials were applied as enzyme carriers [22,23,24]; however, the use of compounds coupling cellulose and PDMAEMA links has not yet been described.

In this connection, this work is aimed to research the interaction between ficin and carboxymethyl cellulose sodium salt and its graft copolymer with *N*,*N*-dimethyl amino ethyl methacrylate (Na-CMC-*g*-PDMAEMA) and the estimation of the interaction effect on enzyme catalytical activity.

## 2. Materials and Methods

### 2.1. Materials

Ficin (F4165) from Ficus latex purchased by Sigma, Burlington, MA, USA was used as the research object. Azocasein and *N*-benzoyl-*DL*-arginine-*p*-nitroanilide (BAPNA) supplied by Sigma Aldrich, Munich, Germany, were used as a hydrolysis substrate in the catalytic activity evaluation assays. Carboxymethyl cellulose sodium salt with *M_w_* ~90 kDa and degree of substitution of 0.7 (Sigma Aldrich, Munich, Germany) was used as a carrier and in the graft copolymer synthesis. These products were utilized without any purification. *N*,*N*-dimethyl amino ethyl methacrylate (Sigma Aldrich, Munich, Germany) was previously distilled in a vacuum (*T_b_*  =  62–65 °C/11 mm Hg; *n*^20^*_D_* = 1.4395) before the carrier synthesis.

### 2.2. Synthesis and Characterization of the Enzyme Carrier

The Na-CMC-*g*-PDMAEMA graft copolymer was obtained by the following procedure: 0.50 g of a carboxymethyl cellulose sodium salt was dissolved in 85 mL of distilled water. Next, 0.05 g of potassium persulfate was dissolved in 5 mL of distilled water, and the potassium persulfate solution was added to the polysaccharide solution. The mixture was degassed by three freeze-thaw cycles; after that, 0.04 g of sodium metabisulfite was added under an argon flow. The reactor was placed in a water bath and kept for 20–30 min at 40 °C. Next, the preliminarily degassed 1.75 mL of *N*,*N*-dimethyl amino ethyl methacrylate was introduced into the reaction mixture under argon flow. The final reaction mixture with a volume of 100 mL was kept at 40 °C for 18 h. The product obtained was isolated by precipitation into acetone, centrifuged, and dried in a vacuum oven to constant weight. The resulting Na-CMC-*g*-PDMAEMA copolymer was purified with ethanol on a Soxhlet extractor. Purification control was carried out spectrophotometrically. The copolymer obtained and grafted PDMAEMA chains were characterized as described in [13] by grafting efficiency and molecular weight determination.

Grafting efficiency (*GE*) was calculated by the following equation:(1)$$GE=\frac{m_{PDMAEMAg}}{m_{PDMAEMAt}}\times 100,$$
where *m*_*PDMAEMAg*_ and *m*_*PDMAEMAt*_ are the mass of grafted PDMAEMA evaluated from FTIR data, g, and theoretical mass of PDMAEMA which can be produced in the polymerization blend, g, respectively.

Sample preparation for determining the molecular weight of the grafted PDMAEMA chains was carried out as follows: 0.5 g of the Na-CMC-*g*-PDMAEMA copolymer was dissolved in the 0.1 N NaNO_2_/0.1 N HCl water solutions, and the resulting blend was incubated for 24 h at room temperature for total destruction of the polysaccharide chain. The blend was deposited into acetone, and the precipitate obtained was filtered off and dried in a vacuum to constant weight.

The determination of PDMAEMA molecular weight was carried out by gel permeation chromatography (GPC). GPC analysis was carried out using the Agilent 1200 Series Chromatography system (Agilent Technologies, Santa-Clara, CA, USA) equipped with an isocratic pump, refractive index detector, and PLmixC column. As the eluent 0.03 M LiCl in *N*-methylpyrrolidone was used, flow rate was 0.5 mL·min^−^^1^. A total of 20 μL of 1 mg·mL^−^^1^ PDMAEMA solution in *N*-methylpyrrolidone was injected in the column at temperature of 50 °C and analyzed with samples of monodispersed polystyrene applied as calibration standards, experimental data were processed by the OpenLab ChemStation Revision B.01.01 software (https://www.agilent.com/en/product/software-informatics/analytical-software-suite/chromatography-data-systems/openlab-chemstation accessed on 20 December 2022).

It was found that grafting efficiency was 31%, *M_w_* of the grafted chains was 11,109, and polydispersity index was 2.4. The lower critical solution temperature (LCST) of the 0.1 % w/v copolymer aqueous solution was determined by hydrodynamic diameter evaluation using Malvern Zetasizer Nano equipment (Malvern Panalytical, Malvern, UK). It was found that LCST value of Na-CMC-*g*-PDMAEMA is higher than 70 °C, allowing their use as ficin carrier in enzyme optimum regions.

### 2.3. Fourier-Transform Infrared Spectroscopy (FTIR)

To prepare samples for analysis, research objects were washed with buffer solutions in D_2_O. Solutions or solid wet samples were placed on the surface of the ZnSe single bounce attenuated total reflection (ATR) accessory attached to an IRAffinity1 FTIR spectrometer (Shimadzu Scientific Instruments, Kyoto, Japan) and thermostated at 25 °C. For a measurement of IR absorbance from 4000 to 600 cm^−^^1^, 128 scans were co-added to attain a resolution of 4 cm^−1^. IR solution 1.10 software (Shimadzu Inc., Tokyo, Japan) was used for acquisition and analysis of the spectra. To obtain the protein spectra from solutions, the buffer spectra were subtracted. Second-derivative spectra were obtained using a five-point window. The band positions obtained from the second derivative were then used as the initial guess for curve fitting of amid I band of the original spectra employing the fitting routine Fityk 0.9.8 (https://fityk.nieto.pl/ accessed on 20 December 2022).

### 2.4. Molecular Docking

The structure preparation of ficin (PDB ID: 4YYW, https://www.rcsb.org/structure/4YYW accessed on 20 December 2022) for docking was performed according to the standard scheme for Autodock Vina (https://sourceforge.net/projects/autodock-vina-1-1-2-64-bit/ accessed on 20 December 2022), described as follows: atoms and atom coordinates of solvent, buffer, and ligands were removed from the input PDB file. The center of the molecule and box parameters were set manually, ensuring that the protease molecule is completely inside the computational space domain [25].

Model of the structures of Na-CMC and Na-CMC-*g*-PDMAEMA copolymer were drawn in the molecular constructor HyperChem (https://hyperchem.software.informer.com accessed on 20 December 2022) and were successively optimized first in the AMBER force field and then quantum-chemically in PM3 (Parametric Method 3). The ligand in docking had maximum conformational freedom: the rotation of functional groups around all single bonds was allowed. Arrangement of charges on Na-CMC-*g*-PDMAEMA copolymer and Na-CMC molecules and their protonation/deprotonation was performed automatically in the MGLTools 1.5.6 package (https://ccsb.scriptps.edu/mgltools/1-5-6 accessed on 20 December 2022).

To obtain in silico results, we applied sequential (cascade, multiple) docking. For Na-CMC and Na-CMC-*g*-PDMAEMA copolymer, five binding sites were successively modeled in the following way. First, the docking was done for the “enzyme–ligand” complex, and then the further model was augmented with another ligand (i.e., three-monomer chain), and so on. In other words, we used the “blind” or cascade docking, which can be described as follows. In the first stage, we calculated the optimal position of the ligand, and then its structure (fixed at the docking site) became an integral part of the target. Thereby this binding site with high affinity to ligand was blocked. Then, we calculated the position of the second ligand, whose molecule (being in its optimal position on the globule) also became a part of the target. In the third stage, both the bound ligands became part of the receptor. The iterative searching and filling of the optimal binding sites on the target surface are repeated until the positions of all five ligands are calculated. This cascade docking is reasonable for modeling the ligand–receptor interactions in the case of an excess number of ligand molecules. Using this procedure, we successively excluded a number of possible binding sites of ligands by step-by-step filling of the available areas on the surface of the protein–ligand complex. In general, cascade docking made it possible to model and analyze, at the atomic level, the binding between the protein and ligands, in excess of ligands.

### 2.5. Ficin Immobilization

Immobilization of ficin on Na-CMC or on the Na-CMC-*g*-PDMAEMA copolymer was carried out by the following procedure: 20 mL of an enzyme solution with a concentration of 2 mg·mL^−^^1^ in 50 mM tri-sodium borate buffer with pH 9.0 was added to 1 g of the polymer and incubated for 2 h at room temperature. After the end of the incubation, the gel-like precipitate was purified from unbound enzymes by dialysis using a cellophane bag, cut-off of 25 kDa, against 400 mL of 50 mM Tris-HCl buffer with pH 7.5. Purification was carried out until the protein was absent in the washing water and was controlled spectrophotometrically on an SF-2000 spectrophotometer (λ = 280 nm), LOMO-Microsystems, Saint Petersburg, Russia. Thus, the transfer of the enzyme from polymer phase to solution is excluded, which makes it possible to carry out the reactions, obtaining a product that is not contaminated with the enzyme.

### 2.6. Protein Content Assay

Protein content in enzyme conjugates with Na-CMC or Na-CMC-*g*-PDMAEMA copolymer was determined by the Lowry method [26] with the following modification: at the first stage of the analysis, the bonds between the modified polysaccharide and the enzyme molecules were broken. For this end, the immobilized enzymes were treated with a 0.7 M solution of K, Na-tartrate prepared with 1 M NaOH at 50 °C for 10 min [27]. The absence of enzyme degradation was controlled spectrophotometrically on a UV-2550PC spectrophotometer (Shimadzu Scientific Instruments, Kyoto, Japan).

### 2.7. Protease Activity Assay

The proteolytic activity assay of the conjugated enzymes was carried out on the azocasein substrate [28]. The experiments were conducted in 50 mM Tris-HCl buffer with pH 7.5 at 37 °C. The substrate concentration was 0.4% w. Briefly, the sample was dissolved in 200 μL of buffer (50 mM Tris-HCl, pH 7.5), mixed with 800 μL of azocasein solution (0.5% in the same buffer), and incubated for 30 min at the temperatures indicated above. Then, 800 μL of 5% trichloroacetic acid (TCA) solution was added, and after 10 min incubation at 4 °C, the precipitated unhydrolyzed azocasein was removed by centrifugation (3 min 13,000 rpm). The supernatant (1200 μL) was mixed with 240 μL of 1 M NaOH solution, and its optical density was measured at 410 nm.

### 2.8. Amidase Activity Assay

A total of 400 μL of BAPNA (1 mg·mL^−1^) and 400 μL of 1 mM *L*-cysteine solution were added to 400 μL of the enzyme solution (in concentration 1 mg·mL^−1^) in 50 mM Tris-HCl buffer (pH 7.5) or to a suspension of 50 mg of the immobilized sample in 400 μL of 50 mM Tris-HCl buffer (pH 7.5). The solution was incubated for 2 h at 37 °C. Then, the reaction was stopped with 800 μL of 1 M HCl. The optical density was measured at a wavelength of 410 nm.

Total protease and total amidase activity were calculated in units (µM of hydrolazed substrate in min) per ml of solution, specific activity for both types of substrate was calculated in units per amount of protein in solution (mg), measured according to the Section 2.6.

### 2.9. Kinetic Properties

The Michaelis-Menten constant *K*_m_ and the maximum reaction rate *V*_max_ of free and immobilized ficin were calculated by the enzymatic assay at a wide range of substrate concentrations (0.1–500.0 μM) under optimal conditions (50 mM Tris-HCl buffer pH 7.5, 37 °C). The apparent *K*_m_ and *V*_max_ values were calculated according to the Michaelis-Menten curve and Lineweaver-Burk double reciprocal models. The enzyme turnover number *k*_cat_ was calculated as Y = Et × *k*_cat_ × X/(*K*_m_ + X), where X is the substrate concentration, Y is enzyme velocity, and Et is the concentration of enzyme catalytic sites. The calculations were performed using GraphPad Prism 6.0 software (https://www.graphpad.com/support/prism-6-updates/ accessed on 20 December).

### 2.10. Statistical Analysis

All the experimental studies were carried out with at least eight repetitions. Statistical processing of the results was carried out using the Stadia 8.0 Professional software package (http://protein.bio.msu.ru/~akula/Podr2~1.htm accessed on 20 December). The statistical significance of differences between the control and experimental values was determined by Student’s *t*-test (at *p* < 0.05), since all indicators were characterized by a normal distribution.

## 3. Results and Discussions

### 3.1. Researching the Interactions between Ficin and Na-CMC and Na-CMC-g-PDMAEMA

To evaluate the possibility of using Na-CMC and Na-CMC-*g*-PDMAEMA as ficin carriers, a molecular docking study was performed. The complexation of ficin with the polymers occurs through different physical interactions, such as hydrogen bonds, electrostatic, van der Waals, and hydrophobic interactions. Applying the molecular docking method, it is possible to clarify the amino acid participating in complexation and determine the type of its interactions. Moreover, affinity calculation allows us to predict the thermodynamical possibility of complex formation between enzyme and carrier. So, molecular docking is a powerful tool for the evaluation of enzyme interaction mechanisms and the prediction of its structural changes.

Like all papain-like proteases, a ficin molecule is folded into two domains. Domain L is mainly α-helical (α-helices LI, LII, LIII). The key feature of the R domain is its antiparallel β-sheet structure [29]. The R domain also contains two helices: RI and RII, both located on the surface of the molecule at opposite ends of the β-sheet structure, which forms the core of this domain [30,31]. The active site of ficin is located on the border of the L and R domains in a V-shaped cleft and is formed by cysteine (Cys25), histidine (His162), asparagine (Asn176), and glutamine residue (Gln19), which are conserved for all papain-like proteases [32].

The interactions between the enzyme and both ligands are thermodynamically allowed and characterized by negative affinity values (Table 1). The topology of the enzyme-ligand complexes is depicted in Figure 1. Both polymers are located in the interfacial cleft of the enzyme globule. Moreover, the first interacting ligand locates directly in the catalytical pocket and the following molecules are near arranged, forming a “cap” on the ficin globule surface (Figure 1). However, Na-CMC-*g*-PDMAEMA has a branched structure and does not fully fit in the elongated slot going beyond it. Hence, this ligand can interact with the surface amino acids placed near ficin’s active site.

A detailed investigation of interactions between ficin and ligands shows that ficin active site amino acids do not form H-bonds with Na-CMC and Na-CMC-*g*-PDMAEMA (Table 1, Figure 2). Hydrogen bonds are mainly formed with amino acids located in disordered regions. However, ficin generates one H-bond with each carrier through amino acid, attributed to the α-helical structure. It is well known that hydrogen bond formation significantly impacts protein structure as compared to other physical interactions due to their higher energy (~5 kkal/mol). So, the above-described processes may affect the α-helix/β-sheet ratio of ficin.

Cys25 and His162, being parts of the ficin active site and being located in the L- and R-domains, respectively, are involved in electrostatic interactions with carriers. Catalytically valuable Gln19 takes part in hydrophobic interactions with Na-CMC. Among the amino acids involved in weak physical interactions with the carriers, which are mainly low-energetic van der Waals and hydrophobic interactions, a significant part belongs to protein secondary structure elements, which also may affect the conformation changing of conjugated ficin. It should be noted that ligands form H-bonds by the carbohydrate skeletons, while the side PDMAEMA chains and carboxy methylene groups involve in other physical interactions. So, it can be concluded that interaction with Na-CMC and Na-CMC-*g*-PDMAEMA may impact the ficin structure

The SEM data also confirm conjugate formation (Figure 3). The Na-CMC has a scaly surface with a clearly defined fibrous structure. The surface of the graft copolymer also has a fibrous structure, containing smooth areas, which appear on the grafting of PDMAEMA to Na-CMC macromolecules. Ficin is characterized by a smooth surface without any pronounced structural formations. The SEM images of ficin conjugates demonstrate a smooth ficin-like surface indicating the formation of the conjugates. However, ficin layers on the conjugates’ surface are non-ideal and contain round-shaped pores or other defects (Figure 3d,e). As can be seen from Figure 3e, ficin forms the double-level layer; the second layer of adsorbed protein is clearly visible in the pore opening.

The FTIR data correlate with the docking results. Figure 4a demonstrates the spectra of Na-CMC and its conjugate with ficin. The FTIR spectrum of Na-CMC contains a number of characteristic absorption bands at 1069 and 1080 and ones at 1327, 1414, and 1593 cm^−1^, corresponding to vibrations of pyranose cycles and dissociated carboxylic, respectively [33,34,35]. The FTIR spectra of the Na-CMC and ficin conjugate contain the same bands; additionally, ficin amid I band appeared as a shoulder on the left side of the 1593 cm^−1^ band.Besides, the significant shifts and shape-changing of the two-mode band describing pyranose cycle vibrations are observed. This also confirms the conjugate formation and involvement of OH-groups and carbon skeleton of the Na-CMC macromolecules in the interactions with the enzyme, which loses the structure of the polysaccharide.

The FTIR spectra of Na-CMC-*g*-PDMAEMA and its conjugate with ficin are represented in Figure 4b. The spectrum of graft-copolymer contains a number of adsorption bands corresponding to vibrations of Na-CMC backbones—pyranose cycles (1043, 1067, and 1152 cm^−1^) and dissociated carboxylic groups (1327, 1389, 1593 cm^−1^), as well as a band at 1721 cm^−1^ describing the C=O stretching of the side PDMAEMA chains [13]. The spectrum of ficin conjugate with graft-copolymer contains the above-mentioned bands, and a new band at 1634 cm^−1^, attributed to ficin Amide I, appeared. Additionally, all characteristic bands in the conjugate spectrum are shifted. This confirms the conjugate formation and the involvement of carboxylic, hydroxylic, and carbonyl groups of Na-CMC-*g*-PDMAEMA in the interaction.

The Amide I band, attributed to protein C=O stretching vibrations, is informative when it comes to evaluating proteins’ secondary structures; moreover, its position in the IR spectrum is sensitive to H-bond formation [36]. The Amide I band of native ficin in solution is located at 1641 cm^–1^ (Figure 4c,d) indicating the presence of hydrated α-helices and β-sheets in ficin globules [37]. The position of the Amide I band for conjugated ficin is slightly shifted to higher wavenumber values (Δν = 4 cm^–1^) confirming the additional H-bond formation involving amino acids of the α-helices. These interactions can affect the elements of the protein’s secondary structure, especially the content of α-helices in ficin globules. These results are also confirmed by the estimation of the α-helices/β-sheets ratio (Table 2). As can be seen from Table 2, ficin dissolution or conjugation leads to the destruction of α-helices. Interacting with Na-CMC-*g*-PDMAEMA, there was an increase in the content of the ficin β-sheets structures, while significant growth in disordered regions was observed for the Na-CMC-conjugated enzyme. It is interesting that many of ficin’s β-sheet amino acids are involved in weak physical interactions with both carriers; however, only in the case of Na-CMC-*g*-PDMAEMA is a sharp increase of protein β-sheet content observed. So, the conjugated ficin conformation deviates from the native one during the interactions with Na-CMC and Na-CMC-*g*-PDMAEMA.

To confirm the interaction mechanism of ficin with Na-CMC and Na-CMC-*g*-PDMAEMA we carried out a series of protein “desorption experiments”, e.g., ficin release from its conjugate in 50 mM Tris-HCl buffer with pH 7.5 and with different additives. To evaluate the contribution of electrostatic interactions in enzyme complexation, ficin desorption was performed in 4–1000 mmol ammonium sulfate, as well as at different pH in the range of 3–11. For the estimation of hydrophobic interactions, the release with 4–500 mmol TritonX100 was conducted, and experiments in the temperature range of 25–80 °C should demonstrate the impact of hydrogen bonds in the complex formation. During the experiments, protein content and proteolytic activity of ficin in precipitate and supernatant were analyzed. As can be seen from the obtained results (Figure 5), H-bonds and electrostatic interactions are the main driving forces of the ficin complexation process. Hydrophobic interactions also occur; however, their contribution is quite low.

When studying the electric field created by the ficin both in pure form and in combination with ligands, it is worth noting that it covers the entire protein complex and approximately repeats the surface profile of the enzyme molecule. A free protein molecule has a generally positive potential due to the exposure of Arg and Lys residues on the surface of the globule, while rare areas of a negative charge are determined by the presence of Asp and Glu residues in them. The appearance of Na-CMC and Na-CMC-*g*-PDMAEMA in the complex does not change this picture (Figure 6).

So, it was shown that the complexation of ficin with Na-CMC and Na-CMC-*g*-PDMAEMA is thermodynamically allowed and is promoted mainly by hydrogen bonds and electrostatic interactions placing polymers near the enzyme active site. The complexation changes the ficin’s secondary structure by decreasing the α-helix content. All these factors may affect the enzyme’s catalytical ability, which will be evaluated in the next subsection of the paper.

### 3.2. Properties of the Conjugated Ficin

As mentioned above (Table 2), the ficin secondary structure undergoes some changes due to interactions with the Na-CMC and Na-CMC-*g*-PDMAEMA carriers. Hence, it is interesting to evaluate the catalytical ability of the conjugated enzyme.

Despite the fact that the Na-CMC-*g*-PDMAEMA takes part in a greater number of weak physical interactions with ficin (as compared to Na-CMC), the protein amount and the protein immobilization yield are practically equal for both carriers. Taking into account the statistical processing of experimental data, the difference between protein content in the studied samples is not significant. This also emphasizes the importance of hydrogen bonds in the formation of conjugates.

As it was mentioned above, enzyme interactions can significantly affect proteolytic activity [22,39]. The ficin activity assay performed with the azocasein substrate use shows that the total proteolytic activity (in U·mL^−1^ of solution) of the immobilized formulations obtained is lower compared to the native ficin (Figure 7b). However, after estimating the specific activity (in U·mg^−1^ of the protein, Figure 7c) of the immobilized ficin, it is clear that its activity is higher than that of the free enzyme. Apparently, the interaction with carriers promotes a more catalytically favorable conformation of ficin globules modulating the active site and increasing the proteolytic activity of ficin. It is well known that activity recovery is an important factor governing the cost of immobilization.

This suggestion is confirmed by the amidase activity assay (Figure 7d,e). It can be seen that the amidase activity of conjugated ficin is higher as compared to that of the native one. The amidase activity was measured with a small substrate BAPNA. The absence of diffusion and steric restrictions (which are characteristic to azocasein hydrolysis) clearly shows the hyperactivation of ficin after interactions with Na-CMC and Na-CMC-*g*-PDMAEMA.

It is interesting to note that, at pH 7.5, casein (pI 4.6), Na-CMC, and Na-CMC-*g*-PDMAEMA have negative charges, while BAPNA has a positive charge. Therefore, it seems very likely that the casein molecules are repulsed from, and the BAPNA molecules are attracted to the matrices of both carriers. This assumption could probably explain the decrease in the total proteolytic activity of ficin and the increase in its total amidase activity. In addition, it is clearly seen from Table 2 that the immobilization of ficin on Na-CMC and Na-CMC-*g*-PDMAEMA leads to various changes in the enzyme secondary structures, which probably explains the fact, that specific protease activity of ficin with Na-CMC is lower than that with Na-CMC-*g*-PDMAEMA, while the reverse is true for the amidase activity.

The possibility of the practical application of immobilized enzymes is significantly determined by their attitude to changing the pH and temperature of the medium, as well as their reusability. Therefore, in the following series of experiments, we evaluated these characteristics of the obtained products. Figure 8a, b represents the dependence of native and immobilized ficin-specific activity on pH and temperature. As can be seen from the data obtained, ficin conjugation does not shift the enzyme pH-optimum, while immobilized formulations retain catalytical activity better. The immobilized enzyme increases its temperature optimum and keeps more percentage of specific activity as compared to the native one. Also, the reusability experiments (Figure 8c) show that the conjugated ficin formulations keep more proteolytic activity during multiple (7 times) testing as compared to the native one. Also, biocatalysts can be completely separated from the reaction mass indicating that stability of the immobilized formulations is achieved.

Generally, the immobilization of enzymes on carriers changes the kinetic parameters of enzymatic catalysis. Therefore, the apparent maximum steady-state rate (*V*_max_), the apparent Michaelis constant (*K*_m_), and the apparent catalytic constant (*k*_cat_) values of the immobilized and free ficin were calculated (Table 3). As one can see from Table 3, immobilization on both carriers decreases the enzyme’s *K*_m_ constant. At the same time, a 3-fold increase in *V*_max_ and *k*_cat_ values was observed for ficin and Na-CMC complex, suggesting that the catalyst’s efficiency is improved. Apparently, this is the consequence of conformational restrictions due to immobilization.

Our study of the thermo- and pH-stability of the native and immobilized ficin resulted in the temporal dependency of proteolytic activity at different temperatures or pH values (Figure 9). These figures indicate that, under our experimental conditions, the ficin conjugated with Na-CMC and Na-CMC-*g*-PDMAEMA recovers a greater part of its activity as compared to the native ficin.

Native ficin dramatically decreases its activity at *T* = 70 °C, while the immobilized forms keep more than 40% of initial proteolytic activity. As expected, interactions of the enzyme with carriers protect them from the negative influence of the environment and promote the retaining of practically valuable properties. However, at *T* = 80 °C all the investigated samples immediately lost their activity. The study of the pH-stability shows that incubation at pH = 7 (which is near the enzyme optimum) provides the gradual loosening of up to 80% of activity for native ficin, while the immobilized forms retain about 40% of the initial proteolytic activity. At pH = 3, 9, and 11, the activity decrease is sharper; however, the immobilized ficin keeps its proteolytic activity better as well.

Storage stability is also important for the practical application of new biocatalyzers. Figure 10 represents storage stability research performed in Tris-HCl buffer with pH = 7.5 at 37 °C for 168 h. As can be seen, immobilized ficin formulations are more stable compared to the native enzyme and retain up to 80% of total protease activity after 168-h incubation.

## 4. Conclusions

According to molecular docking and ficin release experiments in different conditions, it was found that interactions of ficin and carboxymethyl cellulose sodium salt and graft copolymer of carboxymethyl cellulose sodium salt with *N,N*-dimethyl amino ethyl methacrylate occur mainly through the formation of hydrogen bonds and electrostatic interactions, in which amino acids attributed to α-helices of the enzyme globules are involved. This results in changes in ficin that affect the enzyme’s activity. The total activity of the ficin-immobilized formulations is lower as compared to that of the native enzyme. However, the conjugated ficin globules belong to a more favorable catalytical conformation, which is reflected in the enhancement of the ficin’s specific activity. Also, the immobilized formulations obtained are characterized by better storage and thermo- and pH-stability; they are reusable and possess an enhanced thermooptimum. So, the supposed materials can be promising in the capacity of the ficin immobilization matrices for biomedical or biotechnological applications.

## Figures and Tables

**Figure 1 polymers-15-00649-f001:**
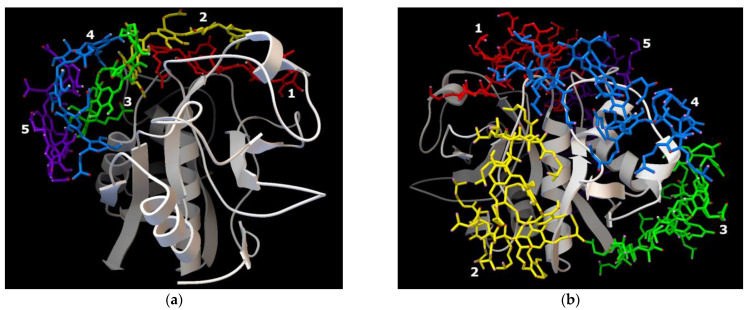
Molecular docking results. The topology of ficin complexes with Na-CMC (**a**) and Na-CMC-*g*-PDMAEMA (**b**).

**Figure 2 polymers-15-00649-f002:**
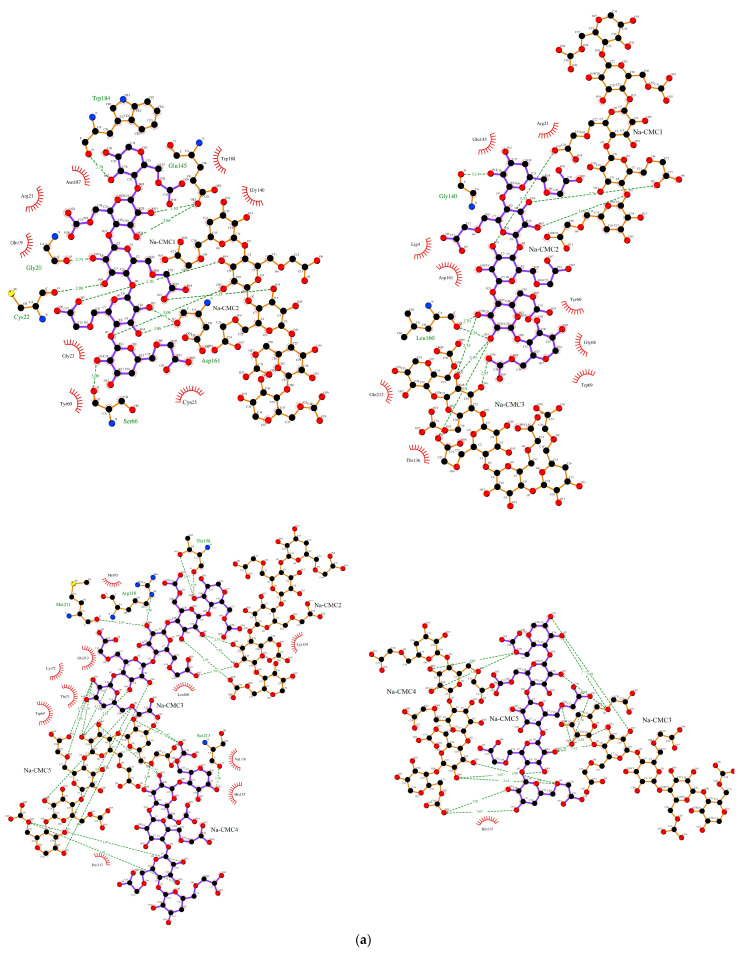
H-bonds (indicated by dashed lines) and interactions between ficin and the molecules of Na-CMC (**a**) and Na-CMC-*g*-PDMAEMA (**b**).

**Figure 3 polymers-15-00649-f003:**
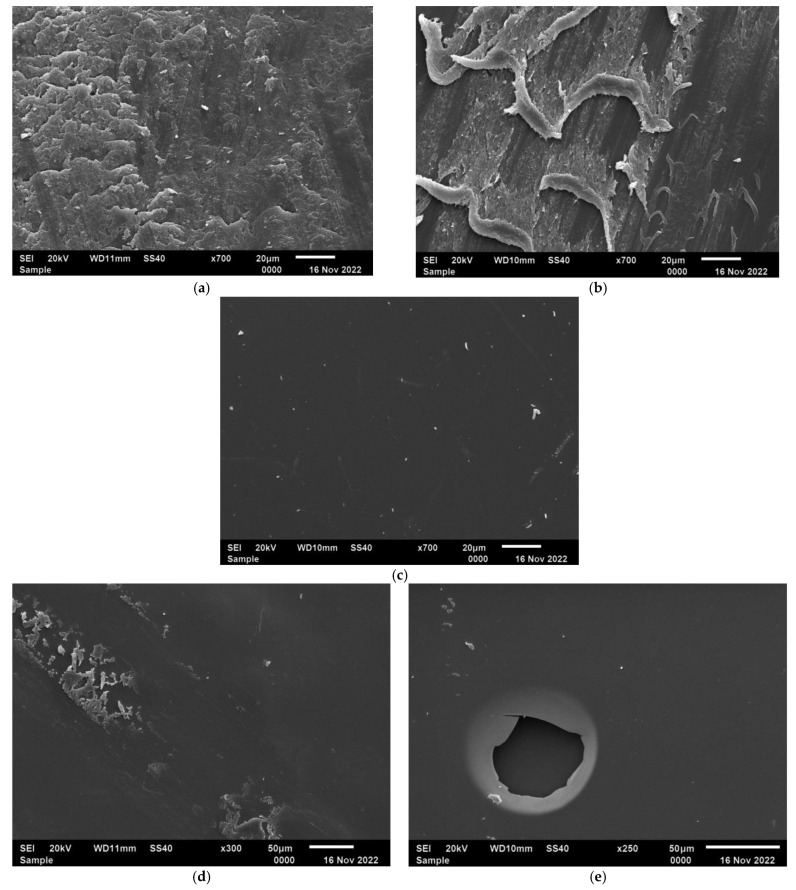
SEM images of surface of Na-CMC (**a**), Na-CMC-*g*-PDMAEMA (**b**), ficin (**c**) and its conjugates with Na-CMC (**d**) and Na-CMC-*g*-PDMAEMA (**e**).

**Figure 4 polymers-15-00649-f004:**
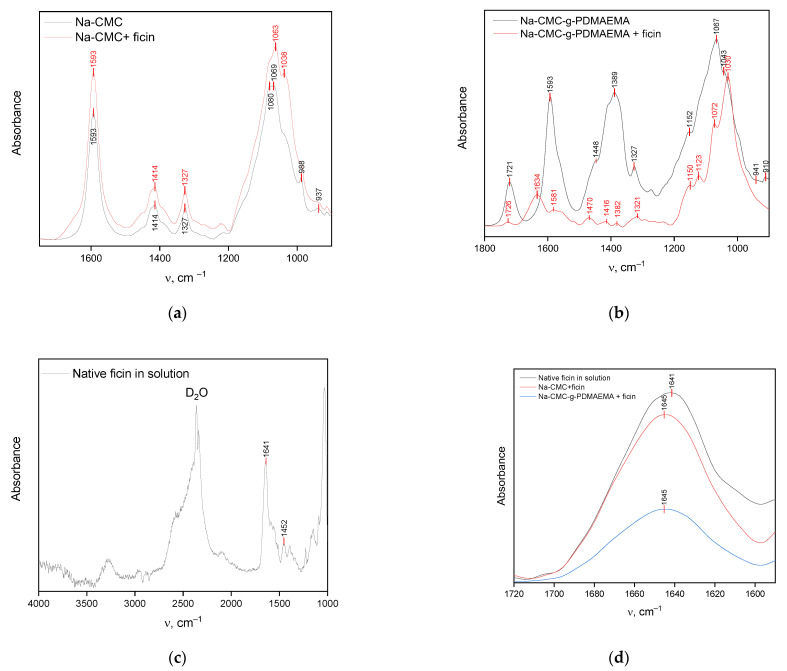
FTIR spectra of Na-CMC (**a**) or Na-CMC-*g*-PDMAEMA (**b**) and their conjugates with ficin; native ficin in D_2_O borate buffer with pH = 9 (**c**); Amide I bands of native and conjugated ficin after subtraction of Na-CMC or Na-CMC-*g*-PDMAEMA spectra (**d**).

**Figure 5 polymers-15-00649-f005:**
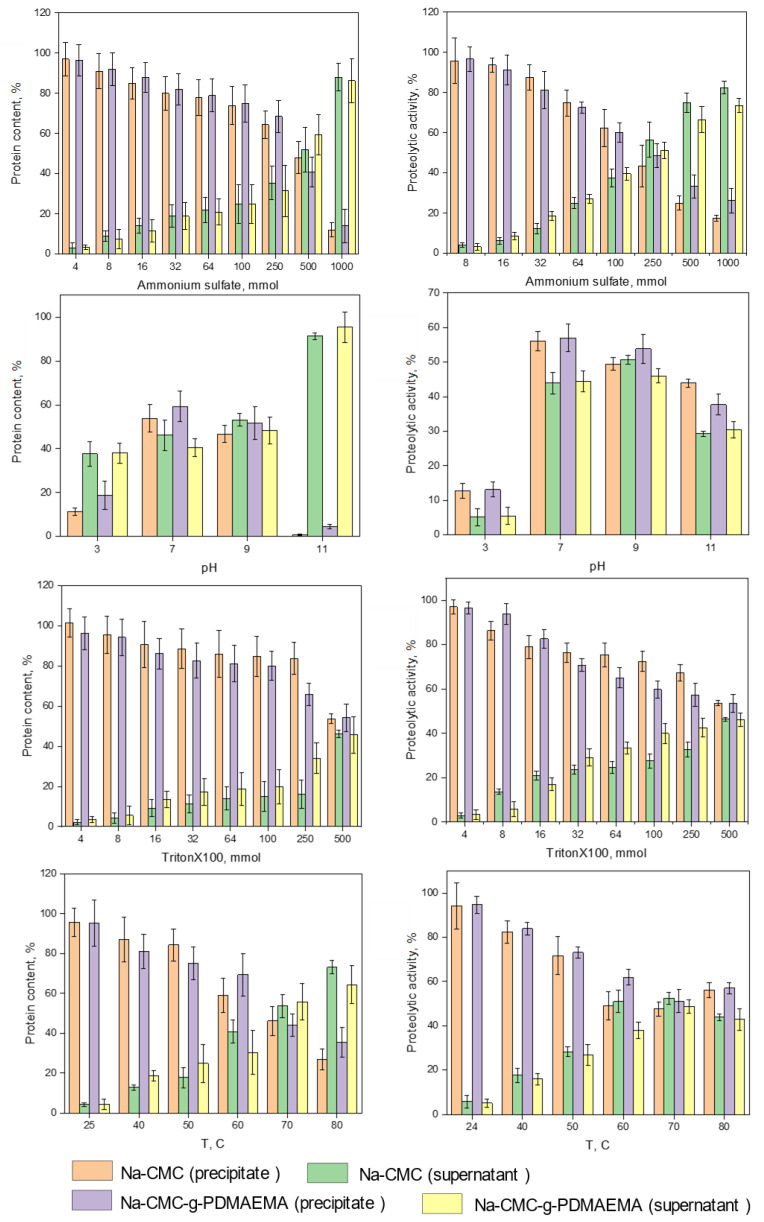
The results of ficin release experiments performed in different conditions.

**Figure 6 polymers-15-00649-f006:**
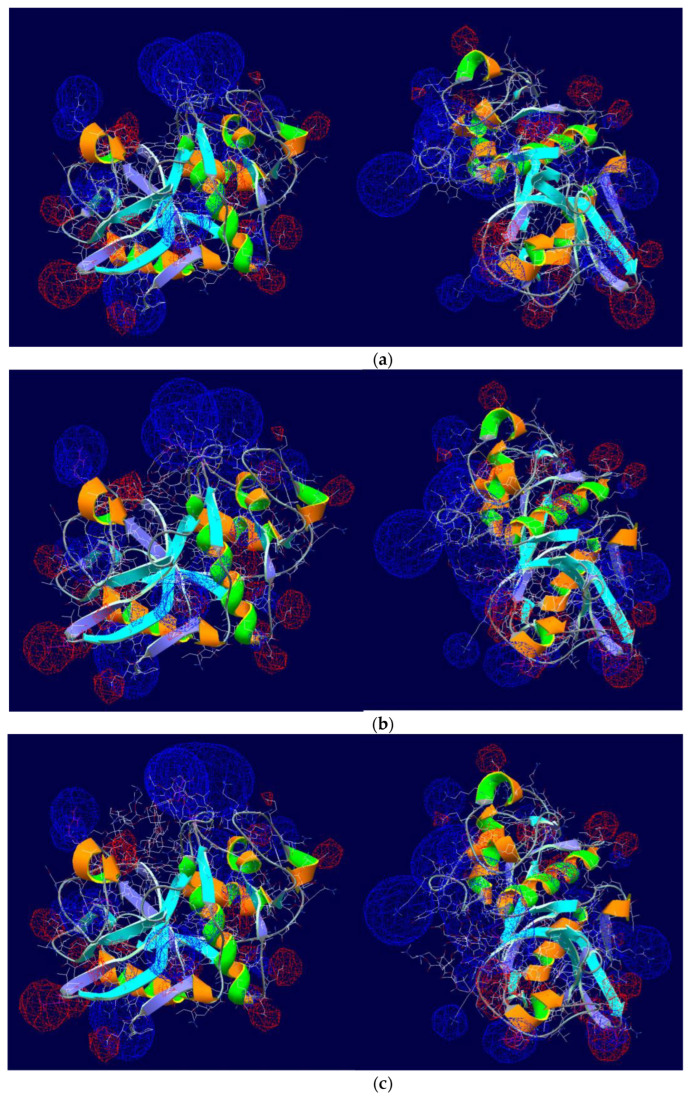
Surface electrostatic potential (in two projections) of ficin molecule (**a**) and complexes of ficin with Na-CMC (**b**) and Na-CMC-*g*-PDMAEMA (**c**): the domains of negative (positive) surface electrostatic potential values are given in red (blue).

**Figure 7 polymers-15-00649-f007:**
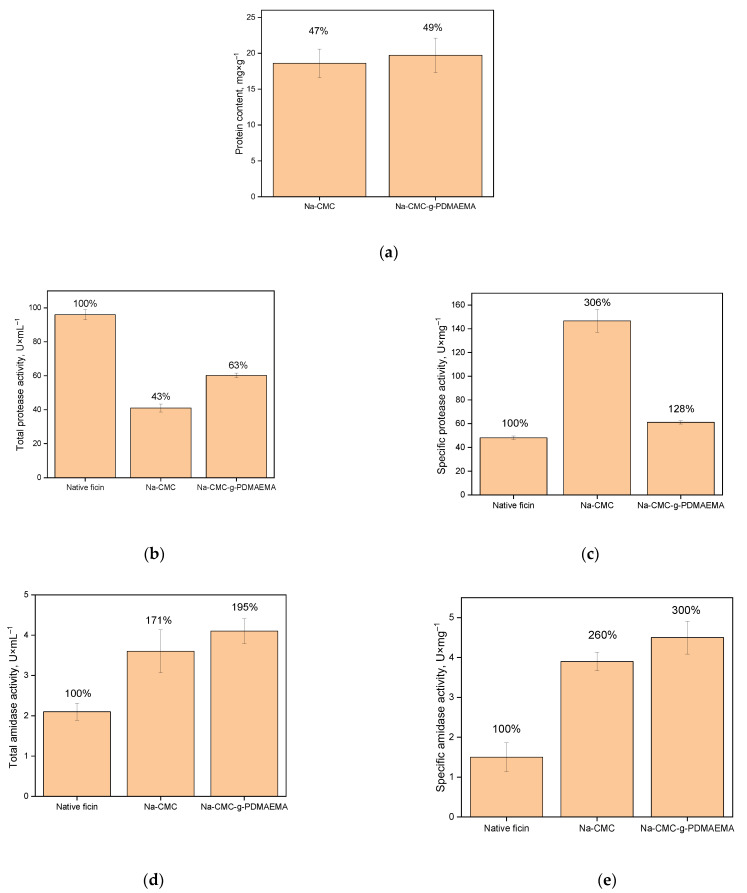
Characteristics of immobilized ficin: (**a**) is protein content (mg·g^−^^1^ of carrier) in the enzyme conjugates with Na-CMC and Na-CMC-*g*-PDMAEMA; (**b**) is total activity (U·mL^−^^1^ of solution) of the enzyme conjugates with Na-CMC and Na-CMC-*g*-PDMAEMA; (**c**) is specific activity (U·mg^−^^1^ of protein) of enzyme conjugates with Na-CMC and Na-CMC-*g*-PDMAEMA; (**d**) is total amidase activity (U·mL^−^^1^ of solution) of the enzyme conjugates with Na-CMC and Na-CMC-*g*-PDMAEMA; (**e**) is specific amidase activity (U·mg^−^^1^ of protein) of the enzyme conjugates with Na-CMC and Na-CMC-*g*-PDMAEMA. The immobilization yield for ficin is expressed as a percentage of the adsorbed enzyme to its amount in solution (**a**), of total activity of immobilized enzyme compared to native enzyme (**b**,**d**), and of specific activity of immobilized enzyme compared to native enzyme (**c**,**e**) (i.g. immobilization recovery) are indicated above bars. All experiments were performed eight times, and the results represent mean ± confidence interval.

**Figure 8 polymers-15-00649-f008:**
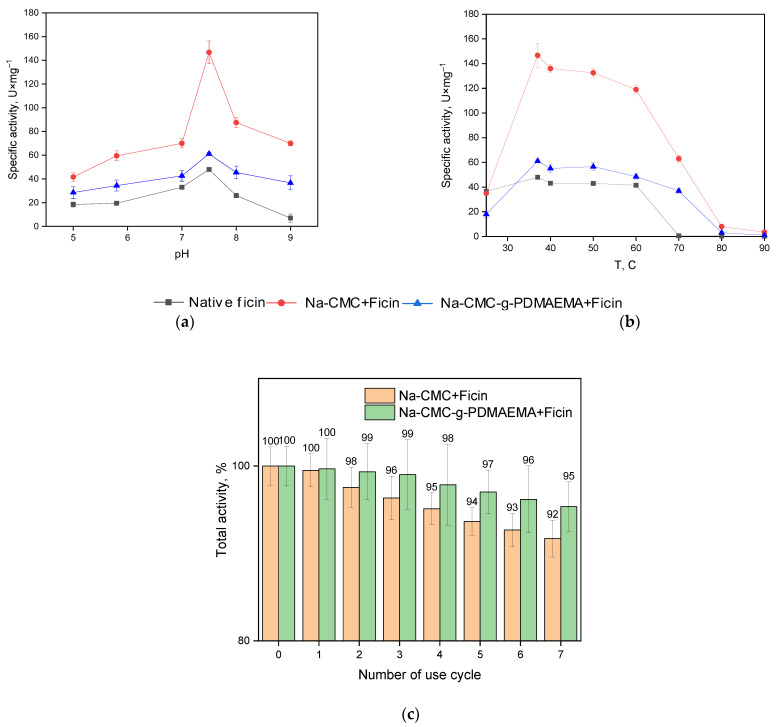
Dependence of ficin specific activity on pH (**a**) and temperature (**b**); the reusability experiments (**c**). Experiments were performed at constant substrate concentration equaled 0.4% w.

**Figure 9 polymers-15-00649-f009:**
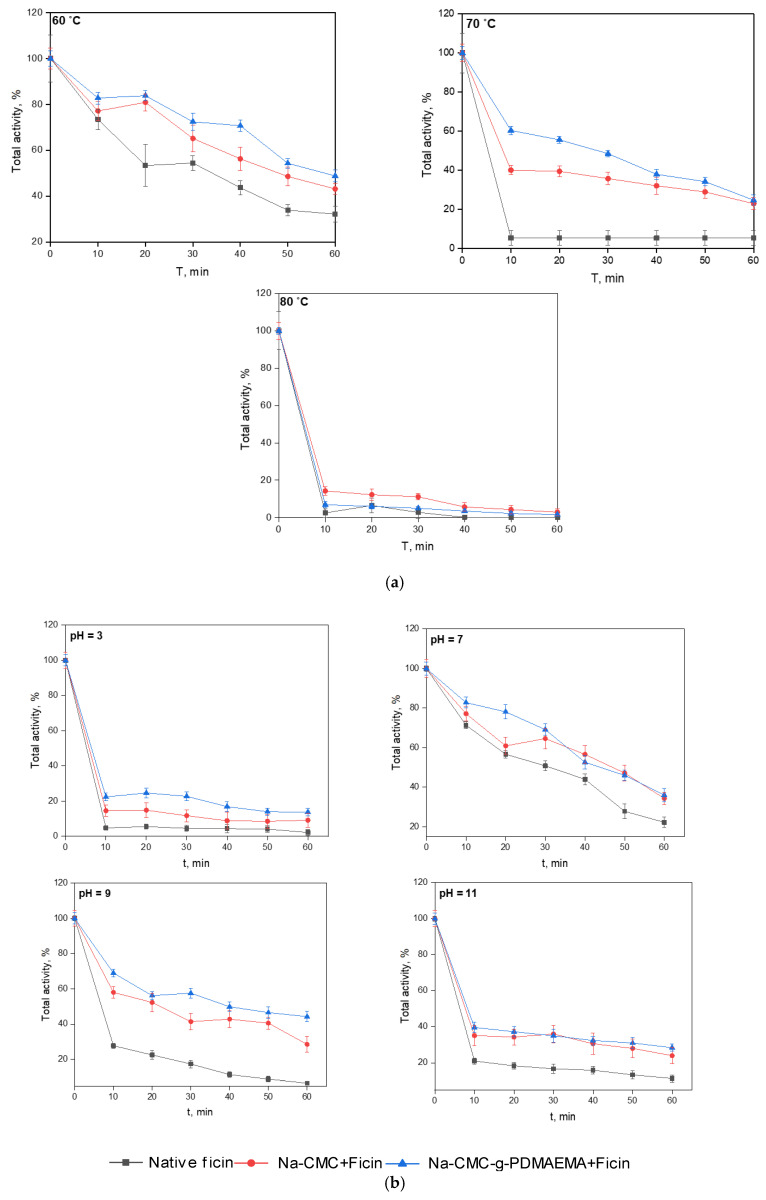
Thermostability (**a**) and pH-stability (**b**) of native and immobilized ficin. Experiments were performed at constant substrate concentration of 0.4% w.

**Figure 10 polymers-15-00649-f010:**
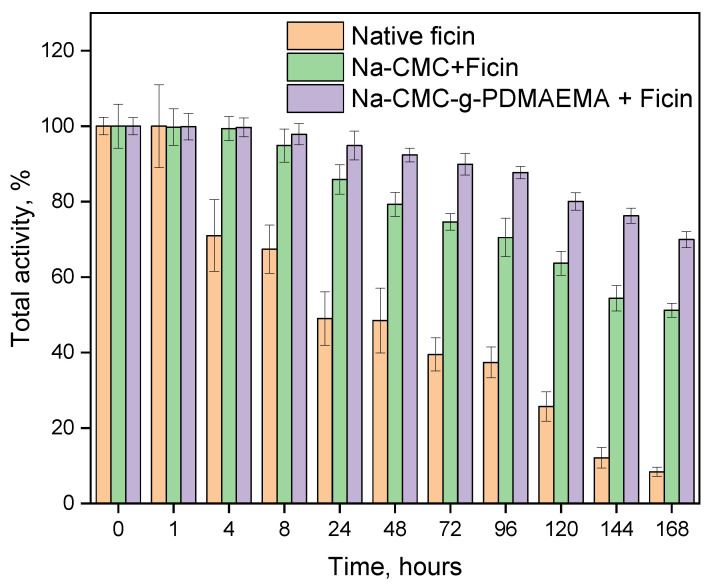
Storage stability of native and immobilized ficin.

**Table 1 polymers-15-00649-t001:** Ficin amino acids interacting with ligands *.

Binding Site Number	Affinity, kcal/mol	Amino Acid Residues Forming	
		H-Bonds, Length, Å	Other Interactions
Amino acids interacting with **Na-CMC**:			
1	−6.7	Gly20, 2.73; Cys22, 2.9; Ser66, 3.09; Glu145 (αR2), 2.66 and 3.10; Asp161, 2.76 and 3.06	**Gln19**, Arg21, Gly23, **Cys25** (αL1), Tyr60, Gly140, Asn187, Trp188
2	−6.5	Gly140, 3.13; Leu160, 2.70 and 3.34	Arg21, Tyr60, Gly68, Trp69, Thr136 (βR), Glu145 (αR2), Asp161, Gln212
3	−6.3	Arg118, 2.96; Thr158, 2.97 and 2.77; Met211, 2.97	Trp69, Met70 (αL3), Thr71 (αL3), Lys72 (αL3), Arg118, Thr158, Lys159, Leu160, Met211, Gln212
4	−6.3	Ser213, 3.16	His115 (βR), Val116, Pro117
5	−6.3	-	His115 (βR)
Amino acids interacting with **Na-CMC-*g*-PDMAEMA**:			
1	−5.0	Cys65, 3.28; Ser66, 2.80; Gly68, 3.25; Asp161, 2.79	Asn18, Arg21, Cys22, Gly23, **Cys25** (αL1), Trp26, Tyr60, Cys65, Ser66, Gly67, Gly68, Trp69, Met70 (αL3), Thr136 (βR), Glu138, Gly140, Gly141, Glu145 (αR2), Leu146, Thr158, Lys159, Leu160, Asp161, **His162** (βR), Trp184, Asn187, Trp188, Gln212
2	−3.7	Asn14, 2.79; Asp43, 2.93; Glu44, 2.10; Leu45, 3.11; Ser47, 3.33;	Arg8, Ile9, Gly11, Val13, Asn14, Pro15, Ile16, Arg17, Asn18, Asp43, Glu44, Leu45, Pro46, Ser47, Gln86, Ser87, Pro90, Tyr91 Val107, Gly185, Thr186, Arg191
3	−3.6	Asn80, 2.79 and 2.76; Ser104, 2.94, 3.01, 3.10 and 3.07; Ile106, 3.16	Ser38 (αL1), Val41 (αL1), Thr42 (αL1), Glu44, Pro46, Ile78 (αL3), Lys79 (αL3), Asn80, Gly81, Lys100 (αR1), Asp101 (αR1), Ser104, Gln105, Ile106, Val107, Ala108, Thr109, Ile110, Asp111
4	−3.7	Thr92, 2.93	Gly20, Arg21, Gln51 (αL2), Asp55 (αL2), Ser87, Asn88, Tyr89, Pro90, Thr92, Ala93, Lys94, Gly96, Glu97, Cys98, Asn99, Lys100 (αR1), Asp101 (αR1), Leu102 (αR1)
5	−3.9	Trp69, 2.81; Val116, 3.13 and 3.28; Arg118, 3.28; Thr71 (αL3), 3.31	Thr58, Ser59, Tyr60, Lys61, Trp69, Thr71 (αL3), Lys72 (αL3), Glu114 (βR), His115 (βR), Val116, Pro117, Arg118, Leu160, Met211, Gln212,Tyr215 (βR)

*—catalytically valuable amino acid residues are bold; protein secondary structure elements are in the brackets.

**Table 2 polymers-15-00649-t002:** The ficin secondary structure.

Structure Elements	Structure Content			
	Ficin X-ray [38]	Ficin in Solution	Ficin with Na-CMC-*g*-PDMAEMA	Ficin with Na-CMC
α-helices	0.27	0.26	0.19	0.15
β-sheets	0.19	0.22	0.46	0.21
Others	0.54	0.52	0.35	0.64

**Table 3 polymers-15-00649-t003:** The kinetic parameters for free and immobilized ficin.

Enzyme Formulations	*K*_m_, μM	*V*_max_, μM mg^−1^ min^−1^	*k*_cat_, min^−1^
Ficin in solution	61 ± 10	1314 ± 215	25 ± 3
Ficin + Na-CMC	19 ± 2	3950 ± 270	77 ± 9
Ficin + Na-CMC-*g*-PDMAEMA	50 ± 8	1580 ± 221	29 ± 3

## Data Availability

Not applicable.

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
