# Peer review of "Carboxymethyl Cellulose-Based Polymers as Promising Matrices for Ficin Immobilization"

_polymers, 2023, doi:10.3390/polym15030649_

Round 1

Reviewer 1 Report (Previous Reviewer 3)

Comments for Authors:

I am aware that this manuscript is a resubmitted manuscript that has been previously reviewed by this referee. Hence, I’ll split up my comments on the changes made by the authors and additional experimental results were added in the latest version.  

(1)   Despite the synthetic approach of Na-CMC-g-PDMAEMA has been reported (reference [13] Polym. Bull. 2021, 78, 2975–2992), the characterization data and methods for the calculation of Mw, grafting efficiency, as well as LCST should be added to the main text and supporting information. By the way, I still can not find the supporting information for this paper. 

(2)   The study of interactions between cysteine protease and copolymer matrices by FTIR, molecular docking, and enzyme activity assay has been significantly improved. However, some minor points still need to be checked or corrected. a) There are five binding sites between matrices and ficin active amino acid residues based on the table 1. However, in figure 1b, binding site 3 is missing in the picture. b) Molecular docking results presented in figure 2, the H-bonds indicated by the green dashed lines are confusing, and the length of the noncovalent bonds is too small for readers. The length of these H-bonds is not such important for the understanding of interactions between cysteine protease and copolymer matrices, which could be removed for the clarity of the presentation of molecular docking results. c) The caption and the format of figure 2 should be checked and revised, such as “Bonds” should be “H-bonds”, and picture b should be moved before the caption. d) The control FTIR spectrum for native ficin in solution has been added, and the discussion for a strict comparison of peak shifts has been improved.

(3)   I highly appreciate the authors have done a systematical enzyme activity study, including pH, temperature, and Michaelis-Menten kinetic study using the copolymer matrices, which offers a convincing point of view to support the novelty of this work. I only have a minor point for the authors consideration. Based on the Michaelis-Menten kinetic parameters included in table 3, the Km of the preset experiments are ranging from 19-61 μM. However, the substrate concentrations used are ranging from 0.1-100 μM. Based on my knowledge of Michaelis-Menten kinetic, 3-5 times higher concentrations should be used in order to obtain accurate numbers of Km, Vmax, and Kcat. A recent paper (J. Am. Chem. Soc. 2021, 143, 13, 5172) including the Michaelis-Menten kinetic study has been listed for consideration.

Overall, the manuscript has been significantly improved and the results presented are sound. I recommend publication after the listed issues have been addressed properly.

Author Response

Reviewer 1

In the following, we response to particular concerns raised by the Reviewer in a step-by-step manner. All our corrections are highlighted in yellow in the text of article.

Comment 1. Despite the synthetic approach of Na-CMC-g-PDMAEMA has been reported (reference [13] Polym. Bull. 202178, 2975–2992), the characterization data and methods for the calculation of Mw, grafting efficiency, as well as LCST should be added to the main text and supporting information. By the way, I still can not find the supporting information for this paper. 

Response: The required information was added to the 2.2 Subsection

Comment 2. The study of interactions between cysteine protease and copolymer matrices by FTIR, molecular docking, and enzyme activity assay has been significantly improved. However, some minor points still need to be checked or corrected.

  1. a) There are five binding sites between matrices and ficin active amino acid residues based on the table 1. However, in figure 1b, binding site 3 is missing in the picture.
  2. b) Molecular docking results presented in figure 2, the H-bonds indicated by the green dashed lines are confusing, and the length of the noncovalent bonds is too small for readers. The length of these H-bonds is not such important for the understanding of interactions between cysteine protease and copolymer matrices, which could be removed for the clarity of the presentation of molecular docking results.
  3. c) The caption and the format of figure 2 should be checked and revised, such as “Bonds” should be “H-bonds”, and picture b should be moved before the caption.
  4. d) The control FTIR spectrum for native ficin in solution has been added, and the discussion for a strict comparison of peak shifts has been improved.

Response: a) Unlike graft-copolymer, Na-CMC interact with ficin globule by several molecules. Therefore only 4 images are shown, however, they contain 5 interacting molecules. Also, file correction may significantly reduce the image quality.

  1. b) The hydrogen bond lengths are the integral part of the output file. File correction may significantly reduce the image quality.
  2. c) Fig. 2’s name was revised
  3. d) Ficin solution spectrum was added to the manuscript

Comment 3. I highly appreciate the authors have done a systematical enzyme activity study, including pH, temperature, and Michaelis-Menten kinetic study using the copolymer matrices, which offers a convincing point of view to support the novelty of this work. I only have a minor point for the authors consideration. Based on the Michaelis-Menten kinetic parameters included in table 3, the Km of the preset experiments are ranging from 19-61 μM. However, the substrate concentrations used are ranging from 0.1-100 μM. Based on my knowledge of Michaelis-Menten kinetic, 3-5 times higher concentrations should be used in order to obtain accurate numbers of Km, Vmax, and Kcat. A recent paper (J. Am. Chem. Soc. 2021143, 13, 5172) including the Michaelis-Menten kinetic study has been listed for consideration.

Response: We regret that this technical typo crept in the text; the correct concentration of substrate at which we conducted the experiment was 0.1–500 μM. We thank the reviewer for their attention. This typo has been corrected in new version of manuscript.

 Thanks for your work!

Reviewer 2 Report (New Reviewer)

In general, this manuscript is well written, sound, clear, and easy to follow. Most of the results are correctly interpreted.

However, there are many points that need to be clarified. Therefore, in my opinion, this manuscript could be published only after major corrections addressing the following questions and comments.

Remarks: *This article is very similar to a previous article by the same authors, cited in reference 15, "Chitosan Graft Copolymers with N-Vinylimidazole as Promising Matrices for Immobilization of Bromelain, Ficin, and Papain" published in "polymers in 2022.

*Most of the keywords are in the title and therefore do not contribute much to the characterization of the content of the paper.

 Comments

L102-103 : how the grafting efficiency and the molar mass of the grafted chains were determined?

L157 : Incubation temperature? Is the gel formation due to the formation of a single complex and/or covalent bonds formation?

Figure 3 : there is no reference to figure 3 in the text. The results of the SEM need to be commented and explained. As it stands, they do not provide any relevant information.

Figure 4 : IR spectra are not easily readable, they must be improved! Furthermore, in Figure 4b, how can be explained the very low intensity of the PDMAEMA carbonyl at 1726 cm-1 in the complex? Idem for the band at 1593cm-1 (carboxylic acid) which is increased in the complex figure 4a. The 2 spectra (with and without complex) of figure 4a are very similar, whereas they are significantly different in figure 4b. Why?

L298: “only in a case of Na-CMC-g-PDMAEMA sharp increase of protein β-sheet con- 298
tent is observed” Is there any explanation for this increase?

L340 : ”ficin structure undergoes changing due to interactions with the Na-CMC and Na-CMC-g-PDMAEMA carriers” . This is not evident in figure 6. could the authors be more precise?

L343-345: The protein amount and the protein immobilization yield are close for both carriers, however, in the case of Na-CMC-g-PDMAEMA, the values are quite higher. In figure7a the values are 47% and 49%. Is the difference really meaningful?

In Figure 7, how do you explain that the specific protease activity of Na-CMC is lower than that of Na-CMC-g-PDMAEMA while the reverse is true for the amidase activity? What are the mechanisms involved?

L371: “Fig. 8c show that conjugated ficin formulations keep about 100 % of proteolytic activity during 7-times testing”. In my opinion figure 8c does not show this assertion: there is a decrease of conjugated ficin formulations.  Moreover, they are no graduations of the total activity in the figure 8c, so it is not possible to evaluate the proportion of decrease!!

Figure 8c shows a rather similar behavior of the Na-CMC-g-PDMAEMA + ficin complex and native ficin but a very different behavior for the Na-CMC +Ficin complex. How can this be explained?

L424-426: “interactions of ficin …. occur mainly through hydrogen bonds formation and electrostatic interactions”. This is probable but not really demonstrated. could this statement be commented on?

Author Response

Reviewer 2

In the following, we response to particular concerns raised by the Reviewer in a step-by-step manner. All our corrections are highlighted in yellow in the text of article.

Comment 1. L102-103: how the grafting efficiency and the molar mass of the grafted chains were determined?

Response: The required information was added to the Subsection 2.2

Comment 2. L157: Incubation temperature? Is the gel formation due to the formation of a single complex and/or covalent bonds formation?

Response: Incubation was performed at room temperature; covalent bond formation can’t occur in these conditions.

Comment 3. Figure 3 : there is no reference to figure 3 in the text. The results of the SEM need to be commented and explained. As it stands, they do not provide any relevant information.

Response: The description was added to the Subsection 3.1 (page 8).

Comment 4. Figure 4: IR spectra are not easily readable, they must be improved! Furthermore, in Figure 4b, how can be explained the very low intensity of the PDMAEMA carbonyl at 1726 cm-1 in the complex? Idem for the band at 1593cm-1 (carboxylic acid) which is increased in the complex figure 4a. The 2 spectra (with and without complex) of figure 4a are very similar, whereas they are significantly different in figure 4b. Why?

Response: High-resolution spectra were added as separate files. The difference in band intensity is due to the difference in normalizing methods. The differences between 4a and 4b spectra are probably due to graft copolymer involved in more efficient interaction with ficin as compared to Na-CMC.

Comment 5. L298: “only in a case of Na-CMC-g-PDMAEMA sharp increase of protein β-sheet content is observed” Is there any explanation for this increase?

Response: Probably, this is due to the influence of PDMAEMA on ficin structure

Comment 6. L340: ”ficin structure undergoes changing due to interactions with the Na-CMC and Na-CMC-g-PDMAEMA carriers” . This is not evident in figure 6. could the authors be more precise?

Response: We have made the following clarification: As mentioned above (Table 2), the ficin secondary structure undergoes changing due to interactions with the Na-CMC and Na-CMC-g-PDMAEMA carriers.

Comment 7. L343-345: The protein amount and the protein immobilization yield are close for both carriers, however, in the case of Na-CMC-g-PDMAEMA, the values are quite higher. In figure7a the values are 47% and 49%. Is the difference really meaningful?

Response: Taking in account statistical processing of experimental data, the difference between protein content in the studied samples is not significant. The text was corrected according to this suggestion.

Comment 8. In Figure 7, how do you explain that the specific protease activity of Na-CMC is lower than that of Na-CMC-g-PDMAEMA while the reverse is true for the amidase activity? What are the mechanisms involved?

Response: This effect is probably caused by electrostatic factors. See the paragraph in lines 408-416 in the revised version.

Comment 9. L371: “Fig. 8c show that conjugated ficin formulations keep about 100 % of proteolytic activity during 7-times testing”. In my opinion figure 8c does not show this assertion: there is a decrease of conjugated ficin formulations.  Moreover, they are no graduations of the total activity in the figure 8c, so it is not possible to evaluate the proportion of decrease!!

Response: Figure 8с with its description has been corrected.

Comment 10. Figure 8c shows a rather similar behavior of the Na-CMC-g-PDMAEMA + ficin complex and native ficin but a very different behavior for the Na-CMC +Ficin complex. How can this be explained?

Response: Probably, interactions with the PDMAEMA grafted chains stabilize the ficin’s protease activity

Comment 11. L424-426: “interactions of ficin …. occur mainly through hydrogen bonds formation and electrostatic interactions”. This is probable but not really demonstrated. could this statement be commented on?

Response: The conclusion has been extended.

Thanks for your work!

Reviewer 3 Report (New Reviewer)

This research reported carboxymethyl cellulose-based polymers for ficin immobilization. The interaction was studied by FT-IR and molecular docking. The specific activities of all ficin samples that immobilized on the polymers are higher than that native enzyme. The weak physical interactions are involved in the activity enhancement. Overall, this is the well-organized manuscript. However, there are still some mistakes and part need to be organized to meet the standard of “Polymers” Journal: Several suggestions/questions are below:

1.      Author chose the carboxymethyl cellulose as the material for study, it’s reasonable, however, author not emphasize the challenge why it usually needs to be modified, especially by using the carboxymethyl sodium salt.

2.      The article lacks a scheme to summarize the main idea, the author needs to add it.

3.      Can the authors examine the CD to determine its α-helical or β-sheet structures to understand the mechanism clearly.

4.      Which kind of the applications may this unique immobilization could be ulitilized?

5.      For do the future immobilized, I suggest the author may independently do the polypeptide with the homo-unit, such as the polyglutamate, poly histidine. Such as, Journal of Controlled Release 2022, 349, 963-982, Biomolecules 2022, 12 (5), 636

Author Response

Reviewer 3

In the following, we response to particular concerns raised by the Reviewer in a step-by-step manner. All our corrections are highlighted in yellow in the text of article.

Comment 1. Author chose the carboxymethyl cellulose as the material for study, it’s reasonable, however, author not emphasize the challenge why it usually needs to be modified, especially by using the carboxymethyl sodium salt.

Response: Carboxymethyl cellulose sodium salt is a cheap industrially-available polymer. Moreover, it is non-immunogenic, biocompatible and biodegradable which make Na-CMC promising material for use in biomedical field. It applies as anti-adhesion gel, tablet filler, drug carrier, etc. The further modification, such as graft copolymerization with high complexing polymers can improve sorption capacity of Na-CMC. Also, incorporating of other functional groups in the Na-CMC macromolecules, new types of interactions with ficin can occur.

Comment 2. The article lacks a scheme to summarize the main idea, the author needs to add it.

Response: The scheme reflecting the work main idea is Graphical Abstract, which was submitted with the manuscript.

Comment 3. Can the authors examine the CD to determine its α-helical or β-sheet structures to understand the mechanism clearly.

Response: Of course, CD is one of the most promising methods to evaluate the protein secondary structure. However, the results providing by the CD method are usually similar with the results obtained by FTIR. The FTIR results are represented in Table 2. Also, we have only 10 days to make corrections in the manuscript, so it’s not enough to carry out suggested experiments. That’s why the CD research will be performed in our further works.

Comment 4. Which kind of the applications may this unique immobilization could be utilized?

Response: As our previously research showed, both free and immobilized ficin has antibacterial and antibiofilm properties. So, the formulations obtained can be applied in wound healing or in cosmetology compounds.

Comment 5. For do the future immobilized, I suggest the author may independently do the polypeptide with the homo-unit, such as the polyglutamate, poly histidine. Such as, Journal of Controlled Release 2022, 349, 963-982, Biomolecules 2022, 12 (5), 636

Response: Thanks for your suggestion. It is promising track for our further research.

Thanks for your work!

Round 2

Reviewer 2 Report (New Reviewer)

The authors have answered the majority of the questions and remarks. Only answers concerning IR spectra are not completely convincing, but this does not prevent the acceptance of this article in its present form.

This manuscript is a resubmission of an earlier submission. The following is a list of the peer review reports and author responses from that submission.

Round 1

Reviewer 1 Report

In short, the study aimed to immobilize a papain-like enzyme, Ficin, into a grafted polymer named Na-CMC-g-PDMAEMA. I don’t think the paper is ready for publication, because the data are just preliminary.

Major comments

1.     Where is the effect of pH and stability and reusability investigations?

2.     The proteolytic activity of immobilized ficin was only determined by azocasein assay. No application of the immobilized ficin was studied.

3.     The characterization was only based on FT-IR. We don’t think it’s sufficient especially to be submitted in Polymers. Other common characterization such as XRD, SEM, DSC, TGA, should be carried out.

Some minor comments

Introduction

1.     Line 33—36 should be revised for better clarity.

2.     Examples of ficin activities should be presented in Introduction (Line 56—66).

Methods and materials

3.     Do you know where Ficin is isolated; the one used in this study?

4.     Abbreviations FTIR and ATR, they have to be defined first.

5.     Considering the big size of the molecule, I don’t think you can get RMSD <2 from redocking protocol. Authors should make a disclaimer about this.

6.     Distribution normality should be determined first before student’s t -test is performed. Please indicate the normality test used in the study. Also provide information about the software used for the statistical analysis.

Results and Discussion

7.     Line 151. “values (Table).” What table?

8.     Please indicate what table in Line 166. Moreover, the presentation is not clear, kindly revise.

9.     Line 152. “catalytical pocket of the enzyme globule”

Reviewer 2 Report

1. Some long and complicated sentences need to be writen into short and concise sentences. eg. The present work is devoted to research of interaction between carboxymethyl cellulose sodium salt and its derivatives named graft copolymer of carboxymethyl cellulose sodium salt and N,N-dimethyl aminoethyl methacrylate with cysteine protease ficin. 

2. what's temperature optimum in sentence "Some of them, for example, ficin, are characterized by the broad substrate specify, pH and temperature optimum"?

3. Line 68-72, the structure is difficult. please rephrase it.

4. What are the definitions for total activity and specific activity? How to measure and how to calculate? And what is the relationship between them?

5. Na-CMC-g-PDMAEMA copolymer  immobilized more ficin. Please explain why  its sepcific activity was much lower than that of Na-CMC? And from your results, Na-CMC was the best, why do you want to graft a N-containing polymer onto CMC? What's your hypothesis and was this against your hypothesis?  

Reviewer 3 Report

Comments for Authors: 

This manuscript submitted by Y. F. Zuev and co-authors reports the synthesis of a graft copolymer (Na-CMC-g-PDMAEMA) and their application as promising matrices for cysteine protease ficin immobilization. Within my limited knowledge of the literature on enzyme immobilization, cellulose-based matrices for enzyme immobilization have been well-developed during the past decades. The presented key result of the study looks relatively interesting and easy to follow the experiments to test their hypothesis. However, this study is lack of enough novelty for the graft copolymers (Na-CMC-g-PDMAEMA) material, since the graft copolymers have been reported and fully characterized by reference [13] (Polym. Bull. 2021, 78, 2975–2992). More importantly, the deep insight into the interaction mechanism of ficin cysteine protease is too preliminary for publication. Most of the direct evidence to support the interaction mechanism is based on superficial analysis, such as FTIR and enzyme activity assay, the molecular insight, deep understanding, and discussions are missing. In addition, the current manuscript is not very well-organized and it is hard to follow the main idea clearly throughout the manuscript. In my point of view, a criticism, related to the previous novelty of the concept, would be to provide sufficient context with more comparisons with previously reported matrices for cysteine protease ficin immobilization to make clear the relevance of the current study. Thus, I can’t recommend this paper for publication in Polymers in the current version. Some concerns and comments for further improving the research have been listed.

(1)   The title of the current manuscript is too general. A possible title may be “Carboxymethyl Cellulose-Based Graft Copolymers as Promising Matrices for Ficin Immobilization”

(2)   As mentioned above, the novelty of the concept, deep molecular insight based on the data presented and the impact on the general scientific community in the frontier research should be included in the abstract and introduction section.

(3)   Some key references for cellulose-based matrices for enzyme immobilization are suggested to add and discuss in the MS. a) Journal of Bioactive and Compatible Polymers. 2016, 31(6):553-567; b) Appl Biochem Biotechnol, 2014, 175:1817-1842; c) Enzym Microb Technol. 2007, 40:1451–1463; etc.

(4)   Page 4, table number (table 1) for ficin amino acids interacting with ligands should be added.

(5)   The conclusion for molecular docking experiments is confusing, and it is difficult to clear the interaction mechanism of Na-CMC-g-PDMAEMA graft copolymer and ficin cysteine protease.

(6)   In Figure 2a, the most marked FTIR peaks of Na-CMC and Na-CMC are the same. No significant shifts were observed. The control FTIR spectrum for Native ficin in Figure 2a is missing, which should be included for a strict comparison.

(7)   For enzyme activity analysis, the significance analysis for each group should be added.

Reviewer 4 Report

Dear Authors, manuscript is interesting, well written and formatted. However some elements need at least major revision. The comments are listed below.

Comments;

1.       The aim of study is misunderstanding compare to title of article Carboxymethyl Cellulose and Its Derivative as Promising Matrices for Ficin Immobilization. Authors should decide, and modify title or aim to be more precise.

2.       Authors performed  a molecular docking study to show possible interaction of carrier variants with active site. However it is not clear why ?. Why authors assumed that this interaction is principal – ficin is a protease not glycohydrolase ? Authors assumed also that Na-CMC is composed from relatively short polyglucan chains whereas average mass of Na-CMC suggest much more longer polymer ?

3.       The immobilization by adsorption strongly depend on properties of carrier and molecular structure of enzyme and environmental conditions pH, temp, salinity etc. The performed study should be a connected with determined enzyme catalytic constant such as Km, Vmax and Kcat. The proper orientation of enzyme molecule on carrier surface is a key of to obtain satisfactory yield, recovery etc. Authors should show how look the electrostatic potential on enzyme surface (with prediction protonation state of carrier and enzyme of course) and discuss how enzyme is oriented on carrier surface and how it affect its properties. Moreover the multilayer effect should be also taken into consideration.

4.       The yield of immobilization and recovery should be reported. Please read; Sheldon RA, van Pelt S. Enzyme immobilization in biocatalysis: why, what and how. Chem Soc Rev. 2013 Aug 7;42(15):6223-35.

5.       Authors should add information how was optimized immobilization process.

6.       The catalytic constants of free and immobilized enzymes should be analyzed and reported.

7.       The operational parameters of immobilized enzyme should be reported, such as reusability optimal pH, pH stability, optimal temperature, temperature stability, storage stability.

8.       Line 192-200. For correct evaluation of changes in proteins secondary structures by FTIR the bands should/must be deconvoluted. How were spectra normalized for comparison ?

9.       Line – 194-195 Authors could check the structure of ficin in CATH (https://www.cathdb.info/browse/sunburst?from_cath_id=3) or PDBsum (https://www.ebi.ac.uk/thornton-srv/databases/cgi-bin/pdbsum/GetPage.pl?pdbcode=4yyw&template=protein.html&r=wiring&l=1&chain=A) databases to avoid mistakes in description. The fully folded ficin does not contains a beta-barrel only two beta-sheet in alpha beta complex.